# Electrochemical Enrichment and Isolation of Electrogenic Bacteria from 0.22 µm Filtrate

**DOI:** 10.3390/microorganisms10102051

**Published:** 2022-10-18

**Authors:** Sota Ihara, Satoshi Wakai, Tomoko Maehara, Akihiro Okamoto

**Affiliations:** 1Graduate School of Life and Environmental Sciences, University of Tsukuba, Ibaraki 305-8577, Japan; 2International Center for Materials Nanoarchitectonics (WPI-MANA), National Institute for Materials Science (NIMS), Ibaraki 305-0044, Japan; 3Institute for Extra-Cutting-Edge Science and Technology Avant-Garde Research (X-Star), Japan Agency for Marine-Earth Science and Technology (JAMSTEC), Kanagawa 237-0061, Japan; 4Graduate School of Chemical Sciences and Engineering, Hokkaido University, Hokkaido 060-8628, Japan

**Keywords:** extracellular electron transfer, interspecies electron transfer, ultramicrobacteria, *Cellulomonas*

## Abstract

Ultramicrobacteria (UMB) that can pass through a 0.22 µm filter are attractive because of their novelty and diversity. However, isolating UMB has been difficult because of their symbiotic or parasitic lifestyles in the environment. Some UMB have extracellular electron transfer (EET)-related genes, suggesting that these symbionts may grow on an electrode surface independently. Here, we attempted to culture from soil samples bacteria that passed through a 0.22 µm filter poised with +0.2 V vs. Ag/AgCl and isolated *Cellulomonas* sp. strain NTE-D12 from the electrochemical reactor. A phylogenetic analysis of the 16S rRNA showed 97.9% similarity to the closest related species, *Cellulomonas* *algicola*, indicating that the strain NTE-D12 is a novel species. Electrochemical and genomic analyses showed that the strain NTE-D12 generated the highest current density compared to that in the three related species, indicating the presence of a unique electron transfer system in the strain. Therefore, the present study provides a new isolation scheme for cultivating and isolating novel UMB potentially with a symbiotic relationship associated with interspecies electron transfer.

## 1. Introduction

The minimum size of prokaryotes such as bacteria and archaea is expected to be 0.2–0.3 μm, and 0.22 μm pore size filters have been used for microbial sterilization. However, Macdonell et al. reported that some *Vibrio*, *Aeromonas*, and *Pseudomonas* species were isolated after 0.22 μm filtration [1]. These bacteria are called ultramicrobacteria (UMB) and are reported to have unique functions and high phylogenetic novelty [2,3]. The phylogenetic diversity of UMB has reported to comprise more than 15% of the bacterial domain, caused a paradigm shift in evolutionary phylogenetics [4,5]. Many reasons for the ultra-micro cell size of bacteria have been proposed, including the lack of unnecessary genes and proteins for niche survival [6,7,8]. In fact, members of candidate phyla radiation (CPR), a highly diversified group of uncultivated bacteria dominant in the 0.22 μm filtrate, have a small genome (often <1 Mb) due to their symbiotic or parasitic lifestyle [4]. Since most of the UMB are in a viable but non-culturable state in conventional culture systems, information on UMB obtained from cultures (e.g., physiological and morphological characteristics) has been limited.

Interspecies electron transfer (IET) is a symbiotic form of metabolism in microorganisms [9,10,11]. In the case of direct IET nanowires that consist of outer-membrane cytochrome or conductive type IV pili (T4P) have highlighted the importance of transferring electrons to other species and driving their metabolism [12]. The genes encoding T4P are conserved in OD1 and WWE3 genomes, which are members of CPR, and the pili formed from ultra-microcells are bound to other cells [13]. Recently, several genes involved in iron reduction and oxidation in the extracellular membrane have been identified in CPR genomes [14]. In addition, several *Vibrio* and *Aeromonas* species possess extracellular electron transfer (EET)-related genes [15,16]. From these reports, we hypothesized that some UMB have EET capability enabling symbiosis associated with IET and can grow on an electrode independently. In the present study, we examined the capability of UMB from a soil sample filtered through a 0.22 μm membrane to grow in an electrochemical reactor. We isolated a bacterial strain and analyzed its molecular phylogenetic, electrochemical, and morphological characteristics under several culture conditions. Unlike previously reported electrochemical enrichment with a relatively large volume of at least 5 mL [17,18], we used a reactor with 200 µL volume that enabled us to compare different medium conditions at once. The present study provides a scheme for electrochemical enriching UMB for the first time.

## 2. Materials and Methods

### 2.1. Medium and Electrolyte Composition

Gifu anaerobic medium (GAM) broth (Nissui Pharmaceutical, Tokyo, Japan), purged with N_2_:CO_2_ (80:20 *v*/*v*) gas for 30 min, was used for the pre-enrichment of anaerobic bacteria in the soil. For this, 10-fold and 100-fold diluted BD Difco^TM^ LB Broth Miller (1/10, 1/100 LB; Becton, Dickinson and Company, Franklin Lakes, NJ, USA); 100-fold diluted BD Difco^TM^ Marine broth 2216 (1/100 MB; Becton, Dickinson and Company, Franklin Lakes, NJ, USA); and 10-fold and 100-fold R2A broth (1/10, 1/100 R2A; Nihon Seiyaku, Tokyo, Japan) were used for electrical enrichment. R2A agar (conc. 1.5%) was used for isolation.

The defined medium (DM), used as an electrolyte for electrochemical measurements as used in other electrogenic bacteria [19,20], had the following composition (L^–1^): NH_4_Cl: 1 g; MgCl_2_·6H_2_O: 0.2 g; CaCl_2_·2H_2_O: 0.08 g; NaHCO_3_: 2.5 g; NaCl: 10 g; and 4-(2-hydroxyethyl)-1-piperazineethanesulfonic acid: 7.2 g (pH 8.1). Glucose: 10 g was used as an electron donor for electrochemical measurements of the DM (DMG). The DM and DMG were purged with N_2_ gas for 30 min.

### 2.2. Electrochemical Enrichment and Isolation of 0.22 μm Filter–Passable Bacteria

Soil was collected from Tomioka, Iwaki, Fukushima Prefecture, Japan (37°21′ N, 141°01′ E). The collected samples were suspended in an appropriate amount of phosphate-buffered saline (PBS), and GAM broth was added for pre-enrichment. After 24 h of incubation at 30 °C, the enrichment culture was filtered through a 0.22 μm PES filter membrane (Millipore Express, Bedford, OH, USA). Subsequently, 20 μL of the filtrate and 180 μL of five different media (1/10 LB, 1/100 LB, 1/100 MB, 1/10 R2A, and 1/100 R2A) were added to the reactor.

Three screen-printed electrode systems were used for electrochemical enrichment (Metrohm DropSens, Oviedo, Spain). A carbon electrode, formed from carbon nanotubes, was used as a working electrode (surface area, 7.07 mm^2^) and counter electrode. Printed Ag/AgCl was used as a reference electrode. All the electrodes were placed at the bottom of the reactor. The working electrode was poised at +0.2 V (vs. Ag/AgCl) throughout electrochemical enrichment at 30 °C for 5 days. Enriched samples were spread on R2A agar under aerobic conditions. A single isolated strain, NTE-D12, was streaked onto fresh agar plates twice to ensure reliable colony purification.

### 2.3. DNA Extraction and 16S rRNA Gene Phylogenetic Analysis

Bacterial cells were lysed according to a previously described protocol [21] and centrifuged at 9100× *g* for 15 min to remove intact cells. The supernatant containing extracted genomic DNA was used for polymerase chain reaction (PCR). The 16S rRNA gene was amplified using TaKaRa Ex Taq (Takara Bio, Kusatsu, Shiga, Japan) DNA polymerase. The primers for PCR were 10 F (5′-GTTTGATCCTGGCTCA-3′) and 1492 R (5′-GGTTACCTTGTTACGACTT-3′). The amplified DNA was purified using the NucleoSpin^®^ Gel and PCR Clean-up (Takara Bio, Kusatsu, Shiga, Japan), followed by sequencing. The sequence data obtained were compiled using BioEdit to construct the full-length 16S rRNA gene sequence [22]. The 16S rRNA gene sequences of type species were collected from the GenBank database; multiple sequence alignment was performed with MUSCLE [23], and a phylogenetic tree was constructed using the neighbor-joining method [24]. Tree topology was evaluated by bootstrapping with 1000 resamplings [25].

### 2.4. Comparison of the Current Production of the Isolated Strain with Related Species and Differential Pulse Voltammetry (DPV) Analysis

Related *Cellulomonas* type strains, including *Cellulomonas algicola* strain NBRC112905^T^, *Cellulomonas fimi* strain NBRC15513^T^, and *Cellulomonas biazotea* strain NBRC12680^T^, were purchased from the NITE Biological Resource Center’s (NBRC) (Kisaradsu, Chiba, Japan) online catalog. The isolated strain NTE-D12 and the three related species were precultured in R2A broth under aerobic conditions. Cultured cells were harvested during the mid-log phase for each species. The cells were centrifuged at 7150× *g* at 4 °C for 10 min. The harvested cells were washed twice with the DM by resuspending, centrifuging the cells, and removing the supernatant. Washed cells were resuspended with DMG for the electrochemical experiment, adjusted to OD_600_ (optical density measured at a wavelength of 600 nm) = 0.5, and purged with N_2_ gas for 30 min. Resuspended cells were added to three screen-printed electrode systems, and chronoamperometry (CA) was measured poised at +200 mV vs. Ag/AgCl for 12 h. 

For DPV measurements, we used a three-electrode reactor with a tin-doped In_2_O_3_ (ITO) electrode as the working electrode (surface area, 3.1 cm^2^) previously used for characterizing *S. oneidensis* MR-1 [26,27]. The working electrode was placed at the bottom of the reactor. A platinum wire and Ag/AgCl saturated KCl were used as the counter and reference electrodes, respectively. DPV was measured under the following conditions: pulse increment, 5.0 mV; pulse amplitude, 50 mV; pulse width, 300 mV; and pulse period, 5.0 s [26,27]. DPV measurement of NTE-D12 was performed before adding cells and 12 and 24 h after adding cells during CA measurement. All the electrochemical measurements were conducted in an anaerobic chamber filled with N_2_ gas and maintained at 30 °C.

### 2.5. Evaluation of Cell Size under Various Pure Culture Conditions Using Scanning Electron Microscopy (SEM) and Filtration

After culturing the cells in R2A for 24 h, aerobically in DMG for 3 days, anaerobically in GAM for 5 days, and electrically in DMG for 24 h, SEM observations were performed. To promote cell adsorption, poly-l-lysine was applied to the ITO electrode, incubated for 40 min, and washed off with sterilized water. The culture medium was smeared onto the coated surface of ITO and incubated for 50 min. Cells were fixed with 2.5% glutaraldehyde for 30 min at 25 °C, followed by washing three times in 0.1 M phosphate buffer for 10 min. This sample was dehydrated with 25%, 50%, 75%, and 99.5% ethanol gradients for 10 min each. The dehydrated samples were exchanged with 100% t-butanol three times and freeze-dried under vacuum. These samples were platinum-coated under vacuum and observed using a Keyence VE-9800 microscope (Keyence, Osaka-shi, Osaka, Japan). To compare cell size, the cells were counted, and the major and minor axes were calculated using ImageJ ver. 1.53 k [28]. Under the same culture conditions as for SEM, the filter-passing capability of pure cultured cells was evaluated to smear these cultures onto R2A agar.

### 2.6. Whole-Genome Sequence of Isolated Strain and Genetic Characterization

Genomic DNA was extracted from the NTE-D12 strain grown aerobically in R2A medium for 24 h using a Quick-DNA HMW MagBead Kit (Zymo Research, Irvine, CA, USA). The concentration of the extracted genomic DNA was measured using a Qubit 2.0 Fluorometer (Thermo Fisher Scientific, Waltham, MA, USA). DNA was sequenced using the Illumina MiSeq (Illumina, San Diego, CA, USA) and the MinION (Oxford Nanopore, Oxford, UK) platforms. For Illumina sequencing, DNA was processed for shotgun library construction using the KAPA Hyper Prep kit for Illumina (KAPA Biosystems, Wilmington, MA, USA) and then sequenced using a MiSeq Reagent Kit v3 (600 cycles) to generate 2 × 300 paired-end reads. For MinION sequencing, an R9.4.1 Flow cell (FLO-MIN106) and Ligation Sequencing Kit (SQK-LSK109) were used according to the manufacturer’s instruction. Summaries of the MiSeq and MinION data were generated using FastQC and NanoStat. Hybrid *de novo* assembly from the MiSeq and MinION data was carried out using Unicycler ver.0.4.7 [29]. The genomes of *C. algicola*, *C. fimi*, and *C. biazotea* were collected from the GenBank database (the accession numbers are GCA_003851725.1, GCA_004306155.1, GCA_000212695.1, respectively). Each genome was annotated using Prokka ver. 1.14.6 [30] and FeGenie ver. 1.0 [14] for comparative genomic analysis. The subcellular location of genes with heme-binding motifs was predicted using Psortb ver. 3.0.3 [31]. We used Orthovenn2 to compare orthologous relationships for the annotated genes in the four strains [32]. Shared orthologous genes among these strains were depicted through a Venn diagram.

## 3. Results

### 3.1. Electrochemical Enrichment and Isolation of 0.22 μm Filter–Passable Bacteria

To grow and enrich 0.22 μm filter–passable bacteria electrically, we poised the electrode potential at +0.2 V and measured microbial current production (*I_c_*), as is done when *Geobacter sulfurreducens*, a representative exoelectrogen for IET, is assigned to potential of EET [11]. Among the five different medium conditions (representative data of duplicates are shown in Figure 1), only one of the 1/100 R2A broths showed a gradual *I_c_* increase to approximately 1.5 mA/m^2^ after 40 h (Figure 1). At the same time, the *I_c_* decreased to about 0.7 mA/m^2^ after 112 h. *I_c_* was sustained until 120 h, and an increase in turbidity was observed, indicating that electrogenic bacteria grew on the poised carbon electrode in one of the 1/100 R2A broths (1/100 R2A-2). The 16S rRNA sequence of the enriched sample showed a single peak, indicating high enrichment of clonal bacteria (Appendix A). *I_c_* did not increase in another broth and one of the 1/100 R2A (1/100 R2A-1) broths.

After spreading the sample on the R2A agar plate, a single colony was consistently formed after five days. The 16S rRNA sequence of the isolates was identical to that of the electrically enriched sample. The 16S rRNA gene of the isolated strain named NTE-D12, is indicative of the *Cellulomonas* clade. The sequence similarities of the 16S rRNA gene of the three closest related species were 97.9%, 97.5%, and 97.5% to the type strains of *C. algicola*, *C. fimi*, and *C. biazotea*, respectively, indicating that NTE-D12 is not only a 0.22 μm filter–passable bacterium with current production capability but also a novel species (Figure 2).

### 3.2. Electrochemical Characterization of the Isolated Strain

Next, we characterized the EET capability of NTE-D12. Current production was measured in the DM with and without glucose at +0.2 V vs. Ag/AgCl and an initial OD_600_ of 0.5. A higher *I_c_* was observed in the presence of glucose than in its absence. A gradual *I_c_* increase (Figure 3A) was immediately observed upon poising the electrode potential into the cell suspension in the electrochemical system. *I_c_* reached 9.0 mA/m^2^ after 12 h, whereas less than 0.4 mA/m^2^ was observed in the absence of glucose, indicating the association of current production capability with glucose oxidation metabolism. In the DPV measurements (Figure 3B) without cells, a specific peak was not observed. Distinct oxidation peak was observed around −0.05 V vs. standard hydrogen electrode (SHE). Cellular coverage on the electrode surface explains the alteration of the DPV profile, and the peak at −0.05 V is most likely associated with the microbial electron transport pathway.

The currents measured in *C. fimi*, *C. algicola*, and NTE-D12 (*n* = 3 for all strains) continued to increase to a maximum of 6.9, 8.1, and 9.0 mA/ m^2,^ respectively; it did not increase significantly after a maximum of 3.9 mA/m^2^ in *C. biazotea* compared to the others. The growth curve showed that, in the R2A medium, *C. biazotea* grew the fastest among the four strains (Appendix A) and was reported to have glucose oxidation capability [33], suggesting that its EET capability was lower than that of the other strains. These results showed that NTE-D12 was the most capable of utilizing glucose as an electron donor for current production among the four strains. Given that these bacterial strains have a glucose oxidation pathway from glycolysis to the TCA cycle, NTE-D12 most likely uses the same pathway to generate current from glucose oxidation. These results demonstrate that the isolated strain NTE-D12 has EET capability and grows on the electrode surface.

### 3.3. Evaluation of Cell Size in Various Culture Conditions

The cells observed under each culture condition are shown in Figure 4A. The projected cellular area and major and minor axes were quantified from the SEM images of cells precultured under four different conditions. Cells were dehydrated under the same conditions, and we compared the size of more than 10 cells. The smallest cell areas were observed after electrochemical culture in DMG, whereas the largest areas were observed after anaerobic culture in GAM. Using the same DMG, the cellular area under aerobic culture conditions was 1.5 times larger than that under electrochemical culture conditions (Figure 4B). The major axis was the largest in the R2A aerobic culture conditions and was twice as large as that in the electrochemical culture conditions, which was the smallest. Interestingly, even under the same culture conditions, cells with approximately 10-fold different areas were observed in GAM, and the standard deviation of all measurements in GAM was larger than that in other culture conditions (Figure 4C). In contrast, cells below 0.22 μm were not observed in these culture conditions. Growth was not observed in these 0.22 μm filtrates on agar plates. These results showed that the cell size of NTE-D12 varies depending on the culture conditions and growth phase, and cultured cells of these four conditions did not pass through a 0.22 μm filter.

### 3.4. Genomic Features of the Isolated Strain and Comparison with Related Species

We determined the whole-genome sequence of strain NTE-D12. The MiSeq sequencing run generated 21,560,786 reads, with a length of 301 bases. The data size corresponds to a coverage depth of over 1000. Despite using data with sufficient coverage, Unicycler assembly using MiSeq data generated 14 contigs. Therefore, long-read sequencing was performed using MinION. The MinION sequencing run generated 280,000 reads, totaling 3,641,335,730 bases after base-calling. Before hybrid assembly, it was downsampled to 83,829,559 bases consisting of 4000 reads, because the data size was too large to perform Unicycler assembly. Hybrid Unicycler assembly was performed using all available MiSeq data and the downsampled MinION read data, resulting in two circular genomes.

The genomic features of NTE-D12 and the three related species are summarized in Table 1, and the features table created using Prokka is shown in Appendix A. The genome of strain NTE-D12 consists of 3,684,286 bp circular chromosomes and a 5386 bp plasmid encoding 3333 protein-coding sequences (CDS), 54 transfer RNAs, 1 transfer–messenger RNA, and 6 ribosomal RNAs. The G–C content was 73.06%. The total genome sequence length and CDS number of NTE-D12 were smaller than those of the three related species. Type IV prepilin-like protein leader peptide-processing enzymes were encoded in the three related species, but none of the genes constituting T4P were encoded. Flavoprotein is important for the EET model in *Listeria monocytogenes* [34], a Gram-positive bacteria similar to *Cellulomonas*. Each genome encodes an electron transfer flavoprotein, succinate dehydrogenase flavoprotein, and flavohemoprotein. Only *C. algicola* encoded the sulfite reductase (NADPH) flavoprotein, whereas only NTE-D12 encoded FixA, B, and X. None of these proteins were predicted to be localized to the cell wall (Appendix A). We could not annotate any quinone biosynthesis genes in NTE-D12, suggesting that other quinone-related species are involved in the electron transport chain in strain NTE-D12 that cannot be annotated using Prokka. Analysis of iron-related genes using FeGenie showed that none of four strains encoded iron-reduction genes. We also detected genes with a heme-binding motif and predicted their subcellular localization (Appendix A). Among these four strains, NTE-D12 had genes with a heme-binding motif predicted to localize most frequently in the cytoplasmic membrane and cell wall. 

Orthologous gene family analysis using Orthovenn2 resulted in 3740 clusters (*C. biazotea*, 3318; *C. fimi*, 3348; *C. algicola*, 3402; strain NTE-D12, 2313 orthologous clusters) (Figure 5). A total of 1936 orthologs were shared by all four strains, with 75 strain-specific gene clusters. Among the four strains, strain NTE-D12 had the highest number of specific gene clusters (50). Interestingly, an electron carrier activity functional gene cluster was specifically annotated in the NTE-D12. This indicated that strain NTE-D12 has a unique electron transfer system compared to that in other related species.

## 4. Discussion

In the present study, we investigated the EET capability of UMB grown on an electrode after passing through a 0.22 μm filter, based on the hypothesis that some UMB have a symbiotic relationship associated with IET. *I_c_* increased only in 1/100 R2A-2 but not in 1/100 R2A-1 though they are the same medium. Since 0.22 μm filterable bacteria are rare in the environment [35], strain NTE-D12 could have also been present as quite a small amount in the preculture with GAM broth and was contingently only introduced into the one 1/100 R2A-2 culture. The isolation of *Cellulomonas* sp. strain NTE-D12, a candidate novel species, from only one among five different electro-culture systems, verified our scheme for enriching highly novel UMB species. The current production of NTE-D12 was similar compared that of *Shewanella oneidensis* strain MR-1 using the same carbon electrode, which is a typical EET-capable bacteria [36]. NTE-D12 generated the highest current among the four related species. *C. fimi* and *C. algicola* also generated current, whereas *C. biazotea* did not. This result indicated NTE-D12 has higher EET capability than the other related species.

The cell size of strain NTE-D12 varied in the four evaluated culture conditions, and after culturing, the cells did not pass through a 0.22 μm filter. Some of the UMB are considered to have an ultra-microcell morphology to resist stress from the surrounding environment [37,38,39]. Although the effect of stress on cell morphology remains controversial, metabolites from other soil-derived bacteria may create a stressful environment for NTE-D12 in GAM culture.

From genomic feature analysis results, iron-reducing genes involved in EET were not detected in the genomes of the analyzed four strains. Flavoprotein, which is considered to be important for EET strategy in Gram-positive bacteria [34], was not predicted to be localized in the cell wall. These results suggest that these four strains capable of current generation have an unknown EET strategy. The EET mechanism of these *Cellulomonas* needs to be investigated in detail in the future. The genome size of strain NTE-D12 is close to that of the *Sphingopyxis alaskensis* strain RB2256 (3.35 Mb), which is relatively larger than the UMB reported so far [40]. Despite having the smallest genome and number of CDSs compared to the three other related species, strain NTE-D12 has the most unique CDSs, including genes predicted to have heme-binding motifs localized at the cell wall and electron carrier activity. Outer-membrane heme-binding proteins have been speculated to participate in EET by typical iron-reducing bacteria and have been previously suggested to be important [19,41]. The fact that there was only one gene cluster with electron transfer activity indicated that this strain has a unique electron transfer system compared to other related species. This unique electron transfer system is suggested to be associated with higher current production capability. Combined with the results of smaller genome size, this strain may have evolved depending on IET with other species in the environment.

## 5. Conclusions

In the present study, a novel *Cellulomonas* sp. strain NTE-D12 was isolated from microbial consortia filtered through a 0.22-μm-pore filter membrane. The isolated strain has higher EET capability, smaller genome size, and unique electron transfer genes compared to related species. This cultivation scheme, a combination of 0.22 μm filtrate and electrochemical culture, can be used to isolate IET-dependent UMB. In the future, this scheme may be applied to isolate or enrich cultures for CPR, which are substantial unculturable phylogenetic groups that account for most of the UMB. These bacteria have not yet been isolated because of their symbiotic relationships with other bacterial species. Because EET-capable microorganisms are extremely diverse [17,18,34,42], it is conceivable that CPR have a symbiotic relationship based on IET. The cultivation scheme established in the present study will enable the isolation of these uncultivated microorganisms and is expected to provide innovative knowledge on microbial symbiosis.

## Figures and Tables

**Figure 1 microorganisms-10-02051-f001:**
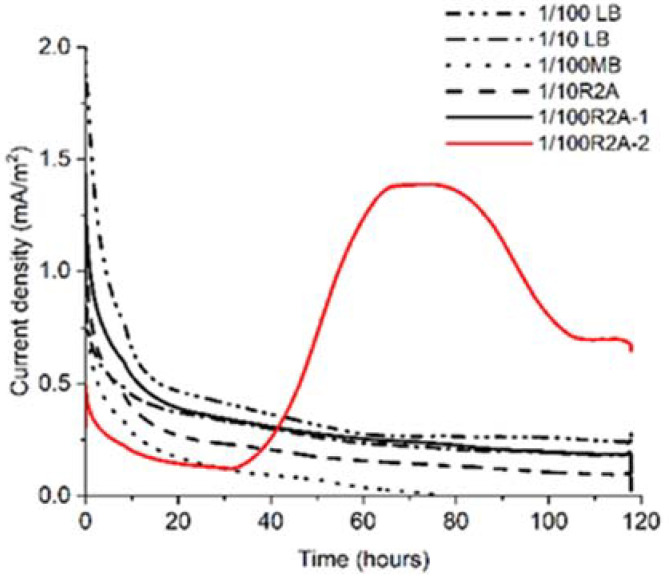
Comparison of current production among five different media for incubating bacteria that can pass 0.22 μm Filter in a soil sample.

**Figure 2 microorganisms-10-02051-f002:**
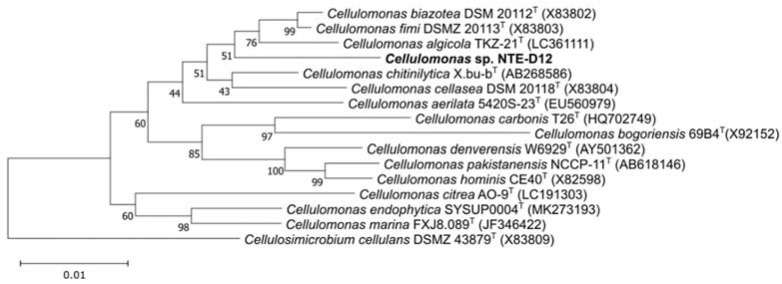
Phylogenetic tree with the novel isolated strain NTE-D12. Numbers at branch points are percentages supported by bootstrap probabilities with 1000 replicates. Bar, 0.01 nucleotide substitution position.

**Figure 3 microorganisms-10-02051-f003:**
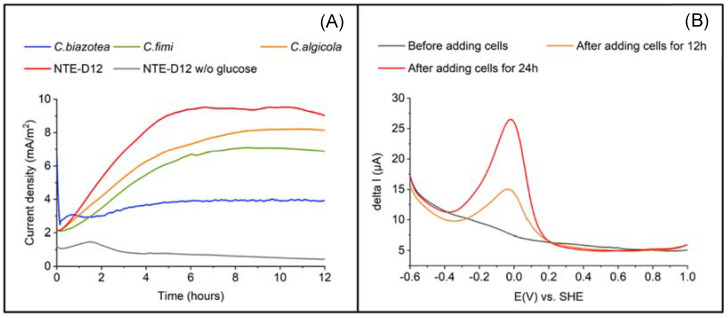
(**A**) Comparison of current production among four strains (*C. biazotea*, *C. fimi*, *C. algicola,* and strain NTE-D12) using carbon electrode as a working electrode. (**B**) Differential pulse voltammograms of strain NTE-D12 before and after adding cells for 12 and 24 h using ITO electrode as a working electrode.

**Figure 4 microorganisms-10-02051-f004:**
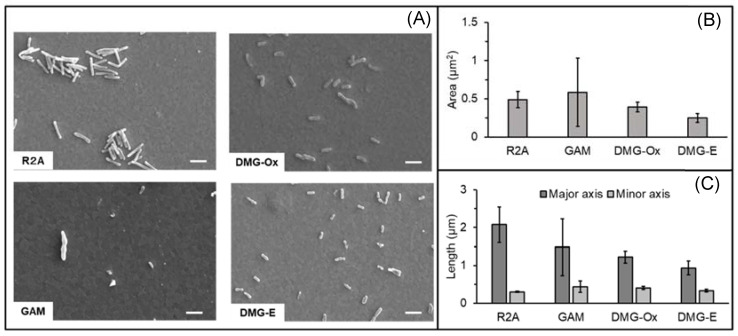
(**A**) Cells observed using scanning electron microscopy (SEM). Scale bar = 1.0 µm. (**B**) Comparison of the observed area and (**C**) major axis and minor axis of cells after aerobic culture in R2A (R2A, *n* = 12), anaerobic culture in GAM (GAM, *n* = 15), aerobic culture in DMG (DMG; DMG-Ox, *n* = 10), and electrochemical culture in DMG (DMG-E, *n* = 18). Error bars represent standard deviation.

**Figure 5 microorganisms-10-02051-f005:**
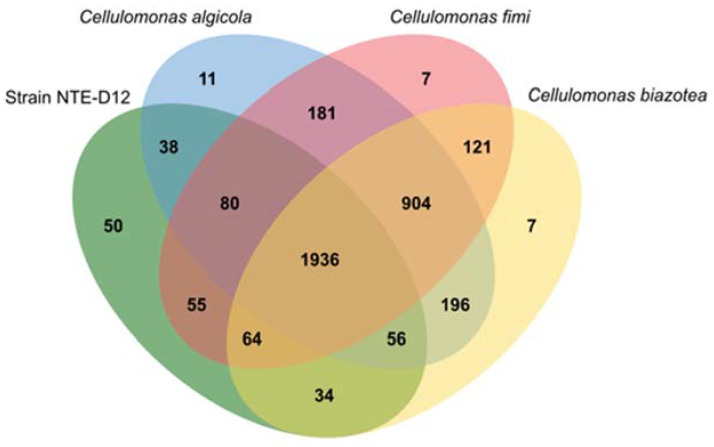
Venn diagram showing orthologues gene clusters among four strains (*Cellulomonas biazotea*, *Cellulomonas fimi*, *Cellulomonas algicola*, and *Cellulomonas* sp. strain NTE-D12).

**Table 1 microorganisms-10-02051-t001:** Genomic features of *Cellulomonas* sp. strain NTE-D12 and three related species.

	*Cellulomonas* sp. Strain NTE-D12	*C. algicola* Strain NBRC112905^T^	*C. fimi* Strain NBRC15513^T^	*C. biazotea* Strain NBRC12680^T^
GenBank accession No.	-	GCA_003851725.1	GCA_000212695.1	GCA_004306155.1
Genome size (Mb)	3.68	4.5	4.27	4.38
CDS number	3333	4047	3812	3911
Iron-reduction gene	N.D.	N.D.	N.D.	N.D.
T4P-related gene	N.D.	1	1	1
Terminal oxidase-related gene	5	6	6	4
Ubiquinone biosynthesis gene	N.D.	5	5	3
Genes with heme-binding motif localized at				
Cytoplasmic membrane	6	4	4	5
Cell wall	1	N.D.	N.D.	N.D.
Flavoprotein localized at				
Cytoplasmic membrane	2	3	1	2
Cell wall	N.D.	N.D.	N.D.	N.D.

## Data Availability

Whole-genome and plasmid sequences were deposited as AP026442 and AP026443, respectively, in the DNA Data Bank of Japan (DDBJ). In addition, the datasets used for the present study, with accession codes DRR392336 and DRR392337, were obtained from NCBI.

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
