# Peer review of "Electrochemical Enrichment and Isolation of Electrogenic Bacteria from 0.22 µm Filtrate"

_microorganisms, 2022, doi:10.3390/microorganisms10102051_

Round 1
Reviewer 1 Report
Ihara et al., reported the isolation of an electroactive ultramicrobacterium. Electrochemical characterization and genomic analyses were performed to suggest the unique extracellular electron transfer system of the isolated strain. Overall, The manuscript was coherently organized and well-written. All the methodologies are sound and clear for me. Essential results are entirely displayed and clearly discussed. It can be accepted by the journal with my following comments addressed:
1) Last two sentences of the introduction: why should the use of micro-reactor be emphasized? I did not see specific discussion in the main text regarding this point. I recommend that both sentences can be removed.
2) Material 2.2: what carbon material was used for carbon electrode? Please clarify.
3) Figure 3B: regarding DPV profile, what did the peak at +0.25 V stand for before the addition of cells? Please clarify.
4) Please check the instruction to see whether citation can be used in ‘conclusion’
Reviewer 2 Report
This study isolated a Cellulomonas sp. strain NTE-D12 from microbial consortia filtered through a 0.22 μm pore filter membrane. The author found that the strain NTE-D12 have higher EET capability, smaller genome size, and unique electron transfer genes compared to related species. This new UMB electrogen is of interest to the field of microbial electrochemistry. However, I still have some comments and questions before it could be accepted for publication.
1. The author should explain more on why they want to isolate UMB electrogens and what we can use these UMB electrogens for.
2. Abbreviations should be defined at the first time they were used.
3. The font size in the Figures was too small. Figure 3A and Figure3B are not in the same height.
4. The cultivation conditions impacted the cell size. Did it also affect t the electricity producing ability?
5. In 2.4., the author mentioned that the cells were resuspended centrifuged by DM, but then wrote “Washed cells were resuspended in DM or DMG”. Please clarify this.
6. In 3.1., the different results between R2A-1 and R2A-2 are contingent? The conclusion that “indicating that the UMB was a minor species in the pre-enrichment condition and only a few species could pass through the 0.22 μm filter” is not rigorous enough, please give more clear interpretations.
Round 2
Reviewer 2 Report
I am happy with the current version.